# 'Seeing' proximal representations: Testing attitudes to the relationship between vision and images

Steven Samuel[1,2]*, Klara Hagspiel[1], Geoff G. Cole[1], Madeline J. Eacott[1]

**1** Department of Psychology, University of Essex, Colchester, United Kingdom, **2** Department of Psychology, University of Plymouth, Plymouth, United Kingdom

* steven.samuel@plymouth.ac.uk

**Data Availability Statement:** All relevant data are within the paper and its Supporting information files.

**Funding:** The authors received no specific funding for this work.

## Abstract

Corrections applied by the visual system, like size constancy, provide us with a coherent and stable perspective from ever-changing retinal images. In the present experiment we investigated how willing adults are to examine their own vision as if it were an uncorrected 2D image, much like a photograph. We showed adult participants two lines on a wall, both of which were the same length but one was closer to the participant and hence appeared visually longer. Despite the instruction to base their judgements on appearance specifically, approximately half of the participants judged the lines to appear the same. When they took a photo of the lines and were asked how long they appeared in the image their responses shifted; now the closer line appeared longer. However, when they were asked again about their own view they reverted to their original response. These results suggest that many adults are resistant to imagining their own vision as if it were a flat image. We also place these results within the context of recent views on visual perspective-taking.

## 1. Introduction

Corrections applied to sensory input by the visual system, such as size constancy, ensure that the retinal image we receive is encoded in a manner that allows us not to be 'fooled' into thinking that a man far away is somehow smaller than a child who is nearby. The distinction between the perception of 'flat plane' retinal image features and perception of an object's 'real' physical features was explicit in Rock's theory of perception [1]. Here, these were referred to as *proximal mode* and *constancy mode* and the distinction was seen as particularly important within developmental psychology [2]. Following the early work of Holway and Boring [3], the standard paradigm required observers to judge the size of different stimuli (usually discs), that were placed at different distances from the viewer, and compare these with a sample stimulus. Based on results from this paradigm, Shallo and Rock [4] put forward what Granrud and Schmechel [2] later called the *proximal mode sensitivity hypothesis*. This suggested that infant perception is more influenced by proximal features than adults. According to Shallo and Rock, as age increases perception is "based on the increased disregard or suppression of proximal mode".

**Competing interests:** The authors have declared that no competing interests exist.

There are however occasions where adult observers need to rely on the retinal size of objects, such as when a visual artist draws a display that faithfully represents what is seen [5]. Here, more distant objects need to be rendered relatively small. Nevertheless, artists also demonstrate difficulty in accessing proximal vision. Perdreau and Cavanagh [6] showed non-artists, art students, and professional artists two cylinders on a computer screen, one presented against a flat background and one in a 3D 'hallway' which gave the illusion of this cylinder being located at a different distance. Their task was to adjust the size of the cylinder in the hallway until the participant judged that it matched the size of the other in terms of their proximal representations. They were told to imagine they were using their fingers to judge the sizes of the items. Results showed that participants consistently overestimated the size of the cylinder presented in the 3D space and moreover that this effect was similar for all groups [for review see 5, see also 7]. Results like these suggest difficulty in or resistance to treating vision (one's own or another's) as a 'flat' image, even with training in representing perception veridically.

In the present experiment we examined whether adults are open to interrogating their own vision in terms of the proximal rather than distal representation. Rather than use a matching-to-standard paradigm, which requires participants to balance the interference effect of distance and size, we investigated whether participants could 'see' the proximal representation of two objects which were already (and explicitly) of equal size but simply located at different distances. This procedure came with the additional benefit that it eliminated the use of 2D depictions of 3D spaces employed in computer-based tasks. In the task, participants stood in a fixed location to the left of two lines that were on the wall of a lab. Fig 1 shows a photo of the lines taken from this location. The lines were the same length but owing to the participant's location the further line would appear shorter in an uncorrected image of their perspective. In the first phase of the experiment we told participants the following: "Both lines are the same length. However, how long does each line actually appear from your visual perspective? Please answer SAME or DIFFERENT." When participants responded 'Different', we then asked participants which line appeared longer. Note that the framing of the question was designed to ensure that participants had to consider their answer in the context of something beyond their own knowledge of the real length of the stimuli. This was made clear by the 'however' following the initial statement of fact. In this way we could examine whether participants would be able and/or willing to access the proximal rather than the distal representation.

Previous research has suggested that training in visual arts offers no significant benefit in the ability to 'see' retinal images [5,6]. We wanted to check whether participants who initially failed to access the proximal representation would be more likely to do so after being shown what such an image could look like. To do so we added two further phases to the task. In Phase 2 we gave the participant a camera and instructed them to take a digital photo of the lines from where they stood. We then asked them the same question about the lengths of the lines but this time based on their appearance in the image. Since the image would clearly depict the closer line as longer (as per Fig 1) we predicted that participants would also judge the same. In Phase 3, we removed the camera and photo and then repeated the question from Phase 1. The photograph will have highlighted the length of the lines in a 2D image, and so those participants who did not judge the closer line to look longer in Phase 1 should now correct their responses and do so in Phase 3.

## 2. Method

We recruited 58 participants from the University of Essex participant pool, each of whom received course credit. Ethical approval was obtained from the University of Essex Ethics Committee Informed consent was recorded in written form and witnessed by the

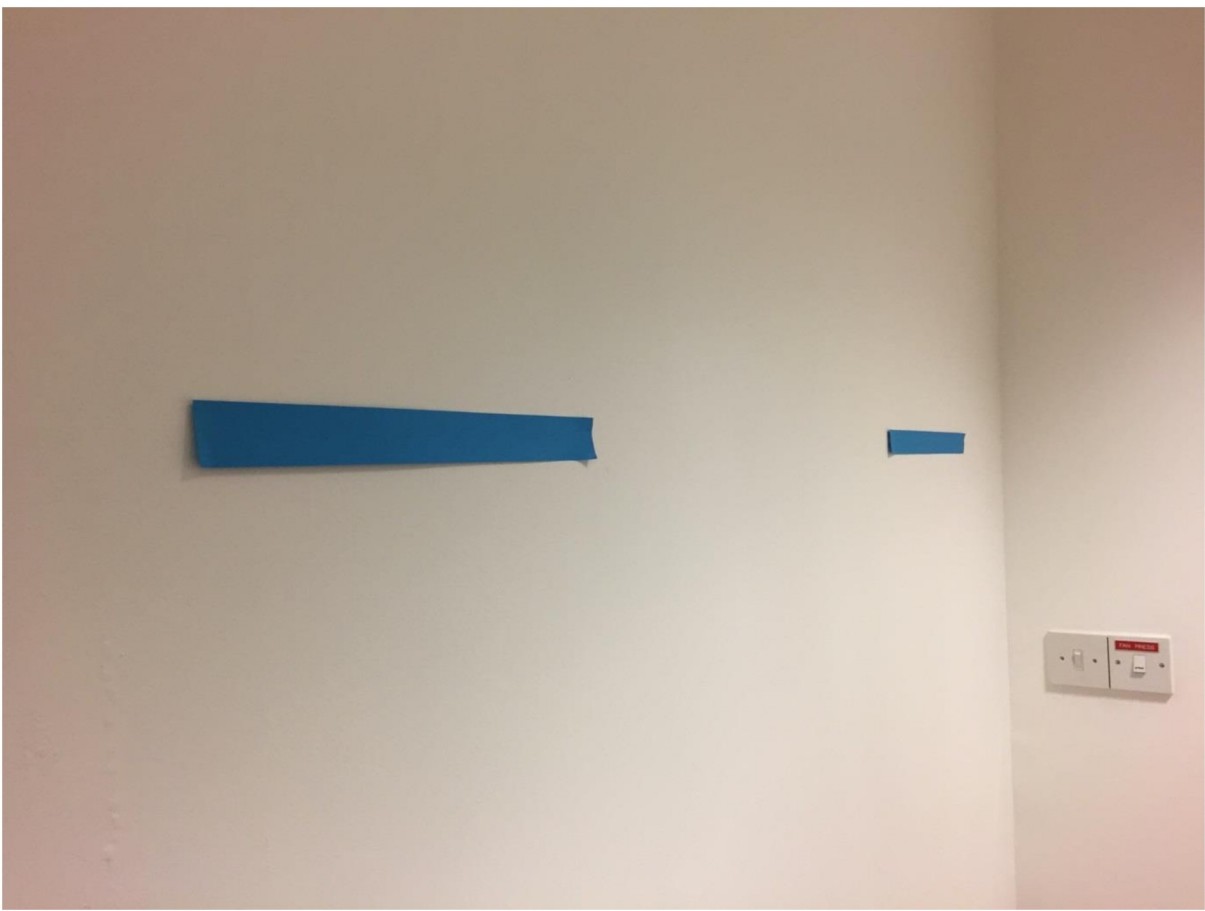

**Fig 1. A photo of the lines on the wall from the participant's location in the room.** The two lines were of equal length but the closer line appeared longer. Note that the photograph is a faithful representation of the lines, i.e. we measured the lines in the photo and found that they corresponded to the real visual angle.

experimenter. Eligibility requirements were normal/correct-to-normal vision and an age between 18–35 ($M$ = 19 years, range 18–24, eight men). Participants stood in a $26cm^2$ square marked on the floor facing a wall 39cm in front of them, but with their head turned to the right so that they could see two lines, each 32cm in length (3cm height), separated by 64 cm. The closest point of the closest line was 32cm from the area of wall in front of the participants. The experimenter stood in the middle of the room. She told the participants that the lines they were looking at were the same length. She then asked them three questions in a fixed sequence (see Table 1).

**Table 1. Scripted instructions to participants.**

| Phase. | Instruction |
|---|---|
| 1 | Both lines are the same length. However, how long does each line actually appear from your visual perspective? Please answer SAME or DIFFERENT. [If DIFFERENT: Which appears longer?]. |
| 2 (Photo) | How long does each line appear in the photo you just took? Please answer SAME or DIFFERENT. [If DIFFERENT: Which appears longer?]. |
| 3 | Now please look at the two lines on the wall once more. How long does each line appear from your perspective? Please answer SAME or DIFFERENT. [If DIFFERENT: Which appears longer?]. |

In Phase 1, we asked them how long the lines actually appeared from their visual perspective, with a fixed-choice answer of either SAME or DIFFERENT, the latter being followed up with an extra question about *which*. If adults can take their own perspective in a visual sense as a result of this appearance question, then more participants should respond that the closest line appeared longer than the converse. In Phase 2 we gave the participant a hand-held digital camera and asked them to take a picture of the two lines from the same location. We then asked them to view the digital photograph they had taken and judge how long the lines appeared in it, using the same question/response structure as before. For Phase 3 the experimenter took back the camera. The participant was then instructed to look at the lines on the wall again, and they were asked the first question once more. Given that everyone had now witnessed a pictorial representation of their own visual perspective in the previous condition, any participants who had not said that the closest line appeared longer in Phase 1 should now be more likely to do so in Phase 3.

## 3. Results

The results are displayed in Table 2.

### Phase 1

We ran a Chi-Square test on the first question concerning whether the lines appeared the same length or different length, with the null hypothesis that each be selected 50% of the time. Overall, 31 judged them the same and 27 judged them different, a contrast that was not significant, chi square $(1, 58) = 0.276$, $p = .599$. However, all 27 participants who responded 'different' judged the closest line to appear longer. Given the absence of any responses favouring the further line as longer, no statistical comparison could be conducted. In sum, approximately half of participants judged the lines to look the same and half judged the closer line to be longer.

### Phase 2

*All* participants said the closest line appeared longer in the photograph. No statistical analyses were conducted since all responses were the same.

### Phase 3

We used the same analysis as for Phase 1. Despite having seen the photograph in Phase 2, results patterned as in Phase 1. Precisely the same number of participants responded 'same' as 'different' (29 each), chi square $(1, 58) = 1$, $p = 1$. Of the 29 who judged them different, 28 responded that the closest line looked longer and only one said the furthest line looked longer, a difference that was significant, chi square $(1, 29) = 25.138$, $p < .001$.

**Table 2. Results.**

|  | Same (%) | Closest line longer (%) | Furthest line longer (%) |
|---|---|---|---|
| Phase 1 | 53 | 47 | 0 |
| Phase 2 | 0 | 100 | 0 |
| Phase 3 | 48 | 50 | 2 |

**Effect of photo on judgements based on vision**

To assess whether seeing the photograph in Phase 2 influenced responses in Phase 3, we dummy coded responses as 0 for 'Closest line longer, 1 for 'Same' and 2 for 'Furthest line longer'. We then ran a Friedman's ANOVA by ranks on all the data. The test found a significant difference in the distribution of responses across the three phases, chi square (2, 58) = 53.118, $p < .001$. Follow-up pairwise comparisons using the Bonferroni correction revealed that more 'closest line longer' responses were made based on the photo (Phase 2) than for both Phase 1 (adj. $p < .001$) and Phase 3 (adj. $p < .001$). Crucially, there was no difference in the frequency of these responses between Phase 1 and 3 (adj. $p = 1$). At the individual level, of the 31 participants who gave 'same' responses in Phase 1, 84% (n = 26) also gave 'same' responses in Phase 3. Five switched from 'same' responses to 'closer line longer' in Phase 3, two switched in the opposite direction, and one switched from 'closer line longer' to 'further line longer.' In sum, seeing the photo in Phase 2 had no significant impact on how participants responded on Phase 3 relative to Phase 1.

## 4. Discussion

The results showed that while about half of participants judged the closer line as being longer, suggesting they did access and use the proximal representation, half did not. However, when basing their judgements on a photo of the lines *every* participant judged the closer line to look longer in a photo they took. Surprisingly, in Phase 3 the vast majority who initially judged the lines to appear the same reverted back to their original response despite having just declared the closer line to appear longer in the photo. This suggests that even when participants are made explicitly aware of what a 2D image of their vision might look like they treated actual sensory input differently. This perseverance suggests considerable resistance to seeing proximal representations of vision.

How do we account for these findings? These results are explicable in terms of what is sometimes called naïve or folk optics, namely folk theories about how vision works. Such beliefs can vary widely from person to person, and lead to very different and often inaccurate responses to the same problems [8,9]. This is nicely exemplified by the *Venus Effect*, whereby an observer sees an agent and a mirror and believes that the agent sees their reflection in the mirror as the observer does, despite the agent and mirror not being along the observer's line of sight [9,10, see also 11]. Importantly for the present experiment, these naïve theories can be inconsistent not only with accepted science but even people's own declarative knowledge [8]. This offers an explanation for why participants proved resistant to changing their responses even after viewing the photograph.

The present work also speaks to the issue of visual perspective taking. A number of people have begun to argue that when humans assume another agent's perspective the representation on which this alternative viewpoint is based is depictive or 'quasi perceptual' [12,13]. Cole and Millett [14], and Cole, Millett [15] have however challenged this claim on theoretical grounds, and in a visual perspective-taking study based on the same stimuli as the present experiment adults also failed to judge that line closer to *another agent* would appear shorter [16]. A depictive representation must, at the very least, code for the relative distances between different points in a scene [17], as seen by the other agent. The present work does not however support the depictive account of perspective taking. Humans seem to be poor at considering even their own perspectives in this way, let alone other people's.

Overall, the results of the present study suggest that while some people are open to interrogating their vision in terms of proximal representations about the same number of people not only have difficulty doing so but also demonstrate considerable resistance to change even after

witnessing what such a representation might look like. It remains an open question as to precisely *why* this should be difficult. One explanation is that people are resistant to the *principle* that vision can be equated to a flat image. According to this view, the reason that those participants who failed to judge the closer line longer is not because they *could not* access a proximal representation but because the logic of the question itself was rejected, as 'corrected' vision is the only type of vision they could reasonably conceive of. This implies that adults are *disinclined* to entertain vision in as a proximal image, even when the context is favourable to such behaviour. However, it does not mean that they are necessarily *unable* to do so.

There is a related but more procedural explanation for the present data that we must consider. Since at least 1960s [18], size constancy researchers have long known that participant interpretation of task requirements is particularly important. Shebilske and Peters [19], for instance, wrote that "the manipulation of instructions is a potent source of variance in constancy experiments. This variance can be attributed to the information given or implied by a particular set of instructions". Our method for circumventing this possibility was to present participants with a photograph (in Phase 2) of their own viewpoint and ensure that responses were close to ceiling. In this Phase *all* participants demonstrated that they could understand and work from a 2D representation of their viewpoint. Thus, while some responses from Phase 1 might be explicable in terms of a misunderstanding of the instruction to judge appearance, by Phase 3 this becomes less plausible.

Another possibility is that those participants who did not judge the closer line to appear longer in Phase 3 may have simply wanted to be consistent in their response when presented with the same question as they had in Phase 1. By this account, our results are not explicable in terms of a disinclination to entertain a proximal image but rather a disinclination to appear inconsistent in an experimental setting. We cannot rule this possibility out, but note that such an account leaves unexplained the fact that by responding differently in Phase 2 and Phase 3 participants also rejected the possibility that their vision corresponded to the *photograph*, which is itself an external and 2D representation of their own vision. A consistency account would thus be more viable if participants maintained the same response that they made in Phase 1 across the rest of the task, including the photograph. Instead, it appears that a disinclination to entertain proximal images provides a better account for our findings.

In sum, these results demonstrate the resistance adults have to entertaining proximal representations of vision. Overall, these results concur with other research, both theoretical and empirical, suggesting that adults do not tend to entertain vision in terms of truly 'depictive' imagery [5,14,15].

## Supporting information

**S1 File.**
(XLSX)

## Author Contributions

**Conceptualization:** Steven Samuel, Geoff G. Cole, Madeline J. Eacott.

**Data curation:** Steven Samuel.

**Formal analysis:** Steven Samuel.

**Investigation:** Klara Hagspiel.

**Methodology:** Steven Samuel.

**Writing – original draft:** Steven Samuel.

**Writing – review & editing:** Steven Samuel, Geoff G. Cole, Madeline J. Eacott.

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
