## [Decision Letter · Decision Letter 0]

30 Jun 2021

PONE-D-21-09371

Can we see our proximal representations?

PLOS ONE

Dear Dr. Samuel,

Thank you for submitting your manuscript to PLOS ONE, and apologies for the slight delay in the review process - this was, in no small part, due to me being on parental leave. In the meantime, your submission has been assessed by two expert reviewers, and you will see that they are both broadly positive about the research question and the quality of the manuscript itself. However, they both present some very useful constructive criticism of the report that would need to be addressed if this were to be published in the journal.

Some of these points pertain to strengthening the write-up itself, with regards to a firmer grounding in the relevant literature and a clearer delineation of some methodological detail. However, the more substantive issue is the requirement for more data - this is something that both reviewers request, and I also concur. One reviewer sees this in terms of the generality of the effect, in the absence of overt manipulation of relevant factors, and the other in terms of how participants interpret the task itself. Both issues are very pertinent, and would usually be relatively simple to address. That said, we are currently in a very atypical situation, and this is one of those paradigms that fundamentally requires face-to-face testing. Still, I do think that it would be appropriate to follow-up on this helpful reviewer feedback, and perhaps it will become more straightforward to do so as the world begins to find its feet again. 

I am, therefore, pleased to invite you to submit a revised version of the manuscript that addresses the points raised during the review process. I do realise that the thought of collecting more data may be disheartening at the moment, but I hope that you appreciate the reasoning here, and I'm confident that it would make for a much better paper if you were able to respond to the commentary you have received. 

Please submit your revised manuscript by Aug 14 2021 11:59PM. Please note that this deadline will have been automatically-generated by the system, and is unlikely to bear in mind the nature of the requests. As such, if you will need more time than this to complete your revisions, please reply to this message or contact the journal office at plosone@plos.org. Please include the following items when submitting your revised manuscript:

We look forward to receiving your revised manuscript.

With kind regards,

Alastair D. Smith

Academic Editor

PLOS ONE

If you are reporting a retrospective study of medical records or archived samples, please ensure that you have discussed whether all data were fully anonymized before you accessed them and/or whether the IRB or ethics committee waived the requirement for informed consent. If patients provided informed written consent to have data from their medical records used in research, please include this information.”

3. Please change "female” or "male" to "woman” or "man" as appropriate, when used as a noun (see for instance https://apastyle.apa.org/style-grammar-guidelines/bias-free-language/gender)

Reviewers' comments:

Reviewer's Responses to Questions

**Comments to the Author**

1. Is the manuscript technically sound, and do the data support the conclusions?

Reviewer #1: Partly

Reviewer #2: No

2. Has the statistical analysis been performed appropriately and rigorously? 

Reviewer #1: Yes

Reviewer #2: Yes

3. Have the authors made all data underlying the findings in their manuscript fully available?

Reviewer #1: Yes

Reviewer #2: Yes

4. Is the manuscript presented in an intelligible fashion and written in standard English?

Reviewer #1: Yes

Reviewer #2: Yes

5. Review Comments to the Author

Reviewer #1: This paper presents a study of size constancy. Participants were asked to access their own vision as if it were an uncorrected 2D image. Approximately half of the participants judged the objects to appear the same, supporting a strong constancy effect. The authors conclude that many adults not only have difficulty interrogating their own perception as if it were a flat image but also show resistance to change.

The topic is interesting, indeed classic, and the analysis is sound. The first problem is that the paper fail to properly acknowledge the large literature and the many studies that already exist. In the introduction it says "relatively few studies have examined the degree to which humans can interrogate their own vision". This is too strong.

A quick list of relevant studies

- Irv Rock used the term "proximal mode" and studied it extensively, including in his book on shape and slant.

- There is work with infants about whether they see the same object at different distances as different objects (e.g., Slater, A., Mattock, A., & Brown, E. (1990). Size constancy at birth: Newborn infants' responses to retinal and real size. Journal of experimental child psychology, 49(2), 314-322)

- Some of the work on mirrors is cited but maybe the most relevant paper is: Lawson, R., Bertamini, M., & Liu, D. (2007). Overestimation of the projected size of objects on the surface of mirrors and windows. Journal of Experimental Psychology: Human Perception and Performance, 33(5), 1027)

The second problem with the paper is the problem we always have with single study papers. That is, what about generality. In this case the problem is even more serious as not only we have a single study, but the design of the study is also minimal. No manipulation of physical size, distance, or angle.

In summary, there are limitations at the moment but on the other hand both of these problems can be addressed. With respect to the literature, more discussing of older papers and rewriting is possible. With respect to the single study, the procedure is simple and it would be easy and informative to replicate and extend the study.

Reviewer #2: The present study tests whether people can access an uncorrected, 2-dimensional representation of their vision (proximal representation), much like a photograph. They show that people perform at chance when judging whether two horizontal lines of equal length appeared to be the same or different lengths from their perspective (where the closer line would appear visually longer in an uncorrected 2D image). Despite all participants then correctly judging a photograph of the lines, they still performed at chance when repeating the first task again. The authors conclude that many people find it difficult to visualise proximal representations, even when presented with an example of how it would look.

I found this manuscript interesting to read and the findings have relevance for the naïve optics literature. The manuscript is well-presented and clearly-written, the methods are clearly described, the data has been made available and the analyses seem appropriate. Unfortunately, however, I have concerns about whether the methodology of the experiment captures exactly what the authors are trying to test. I have particular concerns about how participants were asked to judge the lines and I am not convinced that the task instructions are a) clear enough to be interpreted correctly by participants and b) whether the task can be completed by means other than accessing proximal representations. I therefore recommend major revisions for this manuscript. I would be satisfied if the authors collect more data that convinces me that 1) people do indeed understand the task, 2) the results remain the same, and 3) that the data truly reflects the ability to access proximal representations and cannot be explained by any other means such as access to knowledge.

Major issues:

1. Task instruction

In the task, participants are told: “Both lines are the same length. However, how long does each line actually appear from your visual perspective?”. The authors note that saying “however” encouraged participants to challenge their own knowledge and base responses on proximal representations.

Unfortunately, despite the justification for how the question is framed, I am not convinced that all participants would be able to fully understand what is required of them in this task. It is not clear to me how asking how something “appears” is in reference to how it would look as a flat image. Of course, it is clear in the manuscript that having participants inspect a photo of the lines should indeed clear up any confusion for when they are asked a second time, but I still fail to see how a naive participant would understand the relevance of the photo for the task. This might explain why participants perform at chance and do not change their answers when asked again.

Additionally, by already stating that the lines are of equal length, it sounds like a trick question, and asking participants to repeat the same task again so soon after could result in a general reluctance to change their answer rather than a “resistance to see proximal representations”.

One suggestion would be to instead ask participants a more direct question that really taps into a 2D representation of vision, such as “if you draw the lines as you see them from your perspective, which would you draw longer” or even “if you took a photo…” – this would also make it clear why taking the photo was relevant .

2. Visual representation or knowledge?

The experiment is designed to test whether people have access to a 2-dimensional visual representation of what they can see, however the question can be answered simply by accessing knowledge about size constancy, that closer objects are visually bigger than farther objects, without having to conjure up this image per se. Therefore, even if the participants do understand the question, they do not need to access proximal representations to answer it.

Minor comments:

1. On page 5, line 101, the authors refer to Figure 2 which is not present in the manuscript.

2. In the results for Phase 3 on page 7, line 143-144, the authors incorrectly state that “…the same number of participants responded “same” as “different” (29 each)”. The data actually shows that 28 responded “same” and 30 responded “different”.

3. It would be helpful if the authors could make it clear whether the height of the participant was measured and whether these differences in visual angle would influence how much longer the closer line would “appear” compared to the further line.

4. It would also be helpful if the authors could give more details about how they instructed participants to take the photo – i.e. did they hold the camera exactly where their eyes were or was it held out in front? Did the experimenter inspect the photos that were taken to ensure they were a true depiction of the participant’s perspective?

6. PLOS authors have the option to publish the peer review history of their article (what does this mean?). If published, this will include your full peer review and any attached files.

Reviewer #1: No

Reviewer #2: No

---

## [Author Response · Author response to Decision Letter 0]

14 Jul 2021

PONE-D-21-09371

Can we see our proximal representations?

PLOS ONE

Dear Dr. Samuel,

Thank you for submitting your manuscript to PLOS ONE, and apologies for the slight delay in the review process - this was, in no small part, due to me being on parental leave. In the meantime, your submission has been assessed by two expert reviewers, and you will see that they are both broadly positive about the research question and the quality of the manuscript itself. However, they both present some very useful constructive criticism of the report that would need to be addressed if this were to be published in the journal.

Some of these points pertain to strengthening the write-up itself, with regards to a firmer grounding in the relevant literature and a clearer delineation of some methodological detail. However, the more substantive issue is the requirement for more data - this is something that both reviewers request, and I also concur. One reviewer sees this in terms of the generality of the effect, in the absence of overt manipulation of relevant factors, and the other in terms of how participants interpret the task itself. Both issues are very pertinent, and would usually be relatively simple to address. That said, we are currently in a very atypical situation, and this is one of those paradigms that fundamentally requires face-to-face testing. Still, I do think that it would be appropriate to follow-up on this helpful reviewer feedback, and perhaps it will become more straightforward to do so as the world begins to find its feet again. 

I am, therefore, pleased to invite you to submit a revised version of the manuscript that addresses the points raised during the review process. I do realise that the thought of collecting more data may be disheartening at the moment, but I hope that you appreciate the reasoning here, and I'm confident that it would make for a much better paper if you were able to respond to the commentary you have received. 

Authors general reply:

We are grateful to the Editor and Reviewer for their time and consideration of our manuscript. In the revised document we have highlighted sections where text has been added or edited, for convenience. An overarching theme of our revision concerns the primary research question. In short, we needed to be clearer that we were investigating inclination/propensity rather than ability when it comes to accessing proximal images. It is clear that adults are able to interrogate what they see in such a way that they can develop a reasonably accurate flat-mage version of it. No data is required to prove this point – the fact that still life drawing/painting exists does this job perfectly. Instead, we were more interested in adults’ willingness (or otherwise) to entertain vision in proximal terms. The photo manipulation also makes more sense (we believe) when this point is made more clearly, because the fact that it made no difference to the final, third Phase demonstrates that participants were unmoved: they would not equate their vision to a flat image, even under experimental conditions. As a result, our main approach to our revision has been to ensure that our work is much more clearly situated within this question of willingness/disinclination. We have thus changed the title to reflect this, and removed references to whether adults ‘can’ access proximal imagery, as this word allows the interpretation as ‘ability’ rather than ‘propensity’. 

Reviewers' comments:

Reviewer's Responses to Questions

Comments to the Author

Reviewer #1: This paper presents a study of size constancy. Participants were asked to access their own vision as if it were an uncorrected 2D image. Approximately half of the participants judged the objects to appear the same, supporting a strong constancy effect. The authors conclude that many adults not only have difficulty interrogating their own perception as if it were a flat image but also show resistance to change.

The topic is interesting, indeed classic, and the analysis is sound. The first problem is that the paper fail to properly acknowledge the large literature and the many studies that already exist. In the introduction it says "relatively few studies have examined the degree to which humans can interrogate their own vision". This is too strong.

A quick list of relevant studies

- Irv Rock used the term "proximal mode" and studied it extensively, including in his book on shape and slant.

- There is work with infants about whether they see the same object at different distances as different objects (e.g., Slater, A., Mattock, A., & Brown, E. (1990). Size constancy at birth: Newborn infants' responses to retinal and real size. Journal of experimental child psychology, 49(2), 314-322)

- Some of the work on mirrors is cited but maybe the most relevant paper is: Lawson, R., Bertamini, M., & Liu, D. (2007). Overestimation of the projected size of objects on the surface of mirrors and windows. Journal of Experimental Psychology: Human Perception and Performance, 33(5), 1027)

The second problem with the paper is the problem we always have with single study papers. That is, what about generality. In this case the problem is even more serious as not only we have a single study, but the design of the study is also minimal. No manipulation of physical size, distance, or angle.

Authors reply:

Although we only have one experiment (and thus acknowledge that generality is limited) what is very striking about the results is how huge the effect is. Across the three phases, responses go from chance to ceiling and back to chance. This suggests we are dealing with a basic principle – humans can determine proximal size (Phase 2) but not when they have to introspect their own immediate experience. 

Our study was designed to investigate the tendency (or otherwise) to treat image in terms of proximal representations. We used size constancy as a tool to this end, but please note that the experiment is not a ‘study of size constancy’. Seen this way, the principal characteristic the lines needed to have was to appear different in a proximal image. Phase 2 (the photograph) makes clear that this was the case. If the study had been about size constancy itself, then we agree that it would be useful to manipulate physical aspects of the stimuli.

We have revised the Introduction to reflect the Reviewer’s (very correct) comment regarding developmental research. We have also added the Lawson et al. 92017) reference as suggested in the section on mirrors.

In summary, there are limitations at the moment but on the other hand both of these problems can be addressed. With respect to the literature, more discussing of older papers and rewriting is possible. With respect to the single study, the procedure is simple and it would be easy and informative to replicate and extend the study.

Authors reply:

If it was not for Covid we would certainly have replicated the procedure and extended the work. Given the current rules on testing at Essex and Plymouth, we are not however able to begin this before October at best. It’s not therefore that easy. Given what we said in the previous point about the effect being huge we have opted not to extend the work yet. However, we have added to and amended sections of the Introduction and (more importantly) a large segment of the discussion to explain why we feel that the current data speak quite clearly to the notion of adults’ disinclination to entertain their vision in terms of proximal imagery. 

Reviewer #2: The present study tests whether people can access an uncorrected, 2-dimensional representation of their vision (proximal representation), much like a photograph. They show that people perform at chance when judging whether two horizontal lines of equal length appeared to be the same or different lengths from their perspective (where the closer line would appear visually longer in an uncorrected 2D image). Despite all participants then correctly judging a photograph of the lines, they still performed at chance when repeating the first task again. The authors conclude that many people find it difficult to visualise proximal representations, even when presented with an example of how it would look.

I found this manuscript interesting to read and the findings have relevance for the naïve optics literature. The manuscript is well-presented and clearly-written, the methods are clearly described, the data has been made available and the analyses seem appropriate. Unfortunately, however, I have concerns about whether the methodology of the experiment captures exactly what the authors are trying to test. I have particular concerns about how participants were asked to judge the lines and I am not convinced that the task instructions are a) clear enough to be interpreted correctly by participants and b) whether the task can be completed by means other than accessing proximal representations. I therefore recommend major revisions for this manuscript. I would be satisfied if the authors collect more data that convinces me that 1) people do indeed understand the task, 2) the results remain the same, and 3) that the data truly reflects the ability to access proximal representations and cannot be explained by any other means such as access to knowledge.

Major issues:

1. Task instruction

In the task, participants are told: “Both lines are the same length. However, how long does each line actually appear from your visual perspective?”. The authors note that saying “however” encouraged participants to challenge their own knowledge and base responses on proximal representations.

Unfortunately, despite the justification for how the question is framed, I am not convinced that all participants would be able to fully understand what is required of them in this task. It is not clear to me how asking how something “appears” is in reference to how it would look as a flat image. Of course, it is clear in the manuscript that having participants inspect a photo of the lines should indeed clear up any confusion for when they are asked a second time, but I still fail to see how a naive participant would understand the relevance of the photo for the task. This might explain why participants perform at chance and do not change their answers when asked again.

Additionally, by already stating that the lines are of equal length, it sounds like a trick question, and asking participants to repeat the same task again so soon after could result in a general reluctance to change their answer rather than a “resistance to see proximal representations”.

One suggestion would be to instead ask participants a more direct question that really taps into a 2D representation of vision, such as “if you draw the lines as you see them from your perspective, which would you draw longer” or even “if you took a photo…” – this would also make it clear why taking the photo was relevant .

Authors reply:

We can see that participants might not want to give a different response (on Phase 3) to what they gave just before (on Phase 1) once they fully understand what is being asked of them (after they see the photo). We now discuss this alternative explanation in our revised Discussion. In that section we point out that while the logic of the counterargument is sound insofar as it relates to potential consistency between Phases 1 and 3, the contrast with responses in Phase 2 (the photograph) weakens the explanatory power of this alternative (please see lines 206-227). In relation to the possibility of drawing lines in response to the question, we had considered this (we used this method in our perspective-taking paper on which this paradigm was based), but opted against any method that might allow participants to adopt a strategy whereby they could effectively line their response up with their vision (as a painter holds a brush up in front of their eyes to judge lengths, or a photographer makes a frame with their fingers, etc), as this would be contrary to the spirit of the task as one of mental representation rather than physical action.

2. Visual representation or knowledge?

The experiment is designed to test whether people have access to a 2-dimensional visual representation of what they can see, however the question can be answered simply by accessing knowledge about size constancy, that closer objects are visually bigger than farther objects, without having to conjure up this image per se. Therefore, even if the participants do understand the question, they do not need to access proximal representations to answer it.

Authors reply:

We are in complete agreement here. In fact, we came to this work via our own previous work on visual perspective taking. An opinion is developing which states that VPT is based on a pictorial representation. For this to be true, an observer should be able to determine the relative distance between different points in a display as seen from another individual (i.e., when the observer is attempting to take that person’s perspective). We therefore wanted to know whether an observer can do this with respect to their own proximal vision, let alone someone else’s. We have made the ‘knowledge not pictorial imagery’ point in a number of previous papers (e.g., Cole, Samuel, Millett, & Eacott, 2020). It’s even in the title of Millett, D’Souza, and Cole (2019). However, we note (there as well as here) that geometric reasoning like this is either not applied or applied incorrectly, because as our results show accuracy was at chance.

Minor comments:

1. On page 5, line 101, the authors refer to Figure 2 which is not present in the manuscript.

Authors reply:

We thank the Reviewer for their comments. We have made the relevant correction in the manuscript.

2. In the results for Phase 3 on page 7, line 143-144, the authors incorrectly state that “…the same number of participants responded “same” as “different” (29 each)”. The data actually shows that 28 responded “same” and 30 responded “different”.

Authors reply:

The figures and analysis are correct: 29 responded ‘Same’ and 29 ‘Different’. However, the confusion probably arises because one of the ‘Different’ responders said the further line appear longer, thus 28 gave ‘closer line longer’ and 30 did not.

3. It would be helpful if the authors could make it clear whether the height of the participant was measured and whether these differences in visual angle would influence how much longer the closer line would “appear” compared to the further line.

Authors reply:

We did not measure participants’ height. However, variation in line length would be vanishingly small given the way the lines were arranged from where the participant stood (Figure 1 is a photo taken from precisely that location and the difference is very clear, as all participants verified in Phase 2).

4. It would also be helpful if the authors could give more details about how they instructed participants to take the photo – i.e. did they hold the camera exactly where their eyes were or was it held out in front? Did the experimenter inspect the photos that were taken to ensure they were a true depiction of the participant’s perspective?

Authors reply:

Participants took a photo by looking at the screen on the digital camera, meaning they had to hold the camera in front of their eyes when capturing the image (in the natural manner). The photos themselves were not assessed by the experimenter (indeed, this could have introduced perspective-taking), but given that they saw the participant take the photo and, more importantly, that they showed 100% uniformity in their response based on the photo, we have no concerns.

---

## [Decision Letter · Decision Letter 1]

26 Jul 2021

PONE-D-21-09371R1

‘Seeing’ proximal representations: Testing attitudes to the relationship between vision and images.

PLOS ONE

Dear Dr. Samuel,

Thank you for submitting your manuscript to PLOS ONE. After careful consideration, we feel that it has merit but does not fully meet PLOS ONE’s publication criteria as it currently stands. Therefore, we invite you to submit a revised version of the manuscript that addresses the points raised during the review process.

We look forward to receiving your revised manuscript.

Kind regards,

Julie-Anne Little

Academic Editor

PLOS ONE

Journal Requirements:

Additional Editor Comments:

Thanks for your revisions to the manuscript which the reviewers positively received. One reviewer has a further consideration which you should consider and respond to.

Reviewers' comments:

Reviewer's Responses to Questions

**Comments to the Author**

1. If the authors have adequately addressed your comments raised in a previous round of review and you feel that this manuscript is now acceptable for publication, you may indicate that here to bypass the “Comments to the Author” section, enter your conflict of interest statement in the “Confidential to Editor” section, and submit your "Accept" recommendation.

Reviewer #1: All comments have been addressed

Reviewer #2: All comments have been addressed

2. Is the manuscript technically sound, and do the data support the conclusions?

Reviewer #1: Yes

Reviewer #2: Partly

3. Has the statistical analysis been performed appropriately and rigorously? 

Reviewer #1: Yes

Reviewer #2: Yes

4. Have the authors made all data underlying the findings in their manuscript fully available?

Reviewer #1: Yes

Reviewer #2: Yes

5. Is the manuscript presented in an intelligible fashion and written in standard English?

Reviewer #1: Yes

Reviewer #2: Yes

6. Review Comments to the Author

Reviewer #1: If we were not in a pandemic I would have insisted on a second study. But I take the point about how hard it is to collect data and therefore I agree that this study can be published on its own.

I only have a final comment.

In their reply the authors note that "the experiment is not a study of size constancy", and they also say "we were investigating inclination/propensity rather than ability"

Ok, that is an important point. But in the abstract I read "we investigated whether adults can treat their own vision as if it were an uncorrected 2D image". The word CAN in particular implies that this is a study of the ABILITY to access 2D information despite size constancy. So some more changes to the words is necessary, other readers will read this and conclude that this is a study of size constancy.

Reviewer #2: My initial concerns about this work were (1) whether the task instructions were clear enough for participants to follow as intended, (2) whether the results could be explained by general reluctance to change their answers, rather than a “resistance to see proximal representations”, and (3) whether the task could be completed by accessing knowledge about size constancy, rather than accessing proximal representations.

While my initial request was to collect more data to resolve these issues, the authors have instead provided a comprehensive discussion of these issues in the manuscript, highlighting the limitations they create for this work. I am satisfied with the revisions made to this manuscript.

7. PLOS authors have the option to publish the peer review history of their article (what does this mean?). If published, this will include your full peer review and any attached files.

Reviewer #1: No

Reviewer #2: No

---

## [Author Response · Author response to Decision Letter 1]

27 Jul 2021

Reviewer 1: 

In their reply the authors note that "the experiment is not a study of size constancy", and they also say "we were investigating inclination/propensity rather than ability"

Ok, that is an important point. But in the abstract I read "we investigated whether adults can treat their own vision as if it were an uncorrected 2D image". The word CAN in particular implies that this is a study of the ABILITY to access 2D information despite size constancy. So some more changes to the words is necessary, other readers will read this and conclude that this is a study of size constancy.

Reply:

We agree that the use of 'can' is too ambiguous. We have changed this line in the abstract to read as follows:

In the present experiment we investigated how willing adults are to examine their own vision as if it were an uncorrected 2D image, much like a photograph.

---

## [Editor Report · Decision Letter 2]

12 Aug 2021

‘Seeing’ proximal representations: Testing attitudes to the relationship between vision and images.

PONE-D-21-09371R2

Dear Dr. Samuel,

We’re pleased to inform you that your manuscript has been judged scientifically suitable for publication and will be formally accepted for publication once it meets all outstanding technical requirements.

Kind regards,

Julie-Anne Little

Academic Editor

PLOS ONE

Additional Editor Comments:

Thanks for your response to the recent review, and I am satisfied that the change in abstract wording improves and clarifies the nature of the study for the reader.
---

## [Editor Report · Acceptance letter]

13 Aug 2021

PONE-D-21-09371R2 

‘Seeing’ proximal representations: Testing attitudes to the relationship between vision and images. 

Dear Dr. Samuel:

I'm pleased to inform you that your manuscript has been deemed suitable for publication in PLOS ONE. Congratulations! Your manuscript is now with our production department. 

Kind regards, 

on behalf of

Dr. Julie-Anne Little 

Academic Editor

PLOS ONE